# FlexPro MD®, a Combination of Krill Oil, Astaxanthin and Hyaluronic Acid, Reduces Pain Behavior and Inhibits Inflammatory Response in Monosodium Iodoacetate-Induced Osteoarthritis in Rats

**DOI:** 10.3390/nu12040956

**Published:** 2020-03-30

**Authors:** Min Hee Park, Jae Chul Jung, Stephen Hill, Elizabeth Cartwright, Margaret H. Dohnalek, Min Yu, Hee Joon Jun, Sang Bae Han, Jin Tae Hong, Dong Ju Son

**Affiliations:** 1Division of Life and Pharmaceutical Sciences, Ewha Womans University, 52 Ewhayeodae-gil, Sedaemun-gu, Seoul 03760, Korea; 131pcg01@naver.com; 2R&D Center, Novarex Co., Ltd., 60 Gangni 1-gil, Ochang-eup, Cheongwon-gu, Cheongju, Chungbuk 28126, Korea; jcjung@novarex.co.kr; 3US Nutraceuticals, Inc. d/b/a Valensa International, Eustis, FL 32726, USA; S.Hill@valensa.com (S.H.); E.Cartwright@valensa.com (E.C.); M.Dohnalek@valensa.com (M.H.D.); 4College of Pharmacy and Medical Research Center, Chungbuk National University, 194-21 Osongsaengmyeong 1-ro, Osong-eup, Heungduk-gu, Cheongju, Chungbuk 28160, Korea

**Keywords:** FlexPro MD, krill oil, astaxanthin, hyaluronic acid, inflammation, pain, osteoarthritis

## Abstract

Osteoarthritis (OA) is a degenerative joint disease and a leading cause of adult disability. Since there is no cure for OA and no effective treatment to slow its progression, current pharmacologic treatments, such as analgesics and non-steroidal anti-inflammatory drugs (NSAIDs), only alleviate symptoms, such as pain and inflammation, but do not inhibit the disease process. Moreover, chronic intake of these drugs may result in severe adverse effects. For these reasons, patients have turned to the use of various complementary and alternative approaches, including diverse dietary supplements and nutraceuticals, in an effort to improve symptoms and manage or slow disease progression. The present study was conducted to evaluate the anti-osteoarthritic effects of FlexPro MD^®^ (a mixture of krill oil, astaxanthin, and hyaluronic acid; FP-MD) in a rat model of OA induced by monosodium iodoacetate (MIA). FP-MD significantly ameliorated joint pain and decreased the severity of articular cartilage destruction in rats that received oral supplementation for 7 days prior to MIA administration and for 21 days thereafter. Furthermore, FP-MD treatment significantly reduced serum levels of the articular cartilage degeneration biomarkers cartilage oligomeric matrix protein (COMP) and crosslinked C-telopeptide of type II collagen (CTX-II), and the pro-inflammatory cytokines tumor necrosis factor alpha (TNF-α), interleukin-1β (IL-1β), and interleukin-6 (IL-6), as well as mRNA expression levels of inflammatory mediators, inducible nitric oxide synthase (iNOS) and cyclooxygenase-2 (COX-2), and matrix-degrading enzymes, matrix metalloproteinase (MMP)-2 and MMP-9, in the knee joint tissue. Our findings suggest that FP-MD is a promising dietary supplement for reducing pain, minimizing cartilage damage, and improving functional status in OA, without the disadvantages of previous dietary supplements and medicinal agents, including multiple adverse effects.

## 1. Introduction

Osteoarthritis (OA), also called degenerative joint disease, is a universal debilitating joint disease, and is the most prevalent type of arthritis characterized by synovial inflammation, the gradual loss of articular cartilage and degenerative changes in other surrounding tissues, including the synovium, menisci, ligaments, and subchondral bone, that are caused by multiple risk factors such as age, weight, excessive joint usage, and metabolic or genetic factors [1,2]. OA is the most common joint disorder in the elderly around the world, which can decrease quality of life due to pain, stiffness, loss of function, and disability [3]. Chronic joint pain is the main symptom of OA and is the primary reason for patients to seek medical and pharmacological treatments. Although structural changes, cartilage degeneration, and subchondral bone remodeling are the main contributors to disease progression and joint pain in patients, inflammatory responses mediated by pro-inflammatory cytokines and inflammatory mediators following joint injury also contribute to these pathological events in OA [4,5,6,7,8].

There is no treatment to prevent the development of OA or cure it once it has set in; thus, the goal of OA treatments currently available in clinical practice is mainly aimed at reducing pain, minimizing cartilage damage, and improving or maintain functional status. Commonly recommended treatments include aerobic exercise, strengthening exercises, and medicinal agents, including non-steroidal anti-inflammatory drugs (NSAIDs), corticosteroids, or acetaminophen, which treat OA primarily by reducing pain and inflammation [9]. However, these medications cannot prevent progressive cartilage degradation or repair the damaged cartilage of OA patients, and long-term use of these drugs can lead to various adverse effects, including renal toxicity, gastrointestinal disturbances, diarrhea, nausea, vomiting, or increased cardiovascular risks [10,11,12]. In addition, a recent clinical study reported that chronic but not short-term use of NSAIDs was associated with no clinical improvement in pain and a minimal, non-statistically significant improvement in clinical outcomes in persons with stiffness and functional and structural changes due to OA [13]. Despite their questionable efficacy, prescription NSAIDs are widely used in OA treatment due to the lack of more effective medications.

Therefore, the chronic nature of the disease has encouraged researchers and patients to try various complementary and alternative approaches, including dietary supplements, functional health foods, and nutraceuticals, to relieve pain and improve joint function as well as to manage or slow disease progression [14,15,16,17]. The most widely used and studied dietary supplements or functional health foods for joint pain include those related to chondroprotection, such as glucosamine, chondroitin sulfate, methylsulfonylmethane (MSM), collagen hydrolysates, and hyaluronic acid [18]. However, their effectiveness remains controversial; moreover, they can cause multiple adverse effects, such as stomach upset, constipation, diarrhea, headache, and rash [19,20]. In addition, natural products and nutraceutical products, including curcumin and turmeric extract, green tea extract, resveratrol, citrus fruit extract, *Boswellia serrata*, omega-3 fatty acids, and many others, have been investigated as potential candidates for joint-health promoting dietary supplements [21]. Interestingly, several studies have hypothesized that gut microbiota modulation through the supplementation of specific dietary ingredients including probiotics and prebiotics are able to modify the onset and the progression of OA [22]. Although previous studies have investigated the gut-joint axis in animal model, clinical studies in humans are lacking and no clear mechanism of action has been determined [23]. Therefore, future perspectives should focus on a more detailed understanding of the effects of microbiota modulation in OA patients. Nevertheless, dietary supplements with proven medical benefits, including managing and slowing disease progression and reducing the symptoms of OA, may offer safer alternatives to currently available pharmacological therapies [24,25].

FlexPro MD^®^ (FP-MD), a novel, patented multi-ingredient dietary supplement formulation consisting of krill oil, natural astaxanthin, and sodium hyaluronate, has been shown to markedly reduce pain in subjects suffering from chronic mild-to-moderate knee joint pain [26]. Moreover, another study demonstrated that FP-MD could effectively inhibit lipopolysaccharide (LPS)-induced mRNA expression of pro-inflammatory cytokines and inflammatory markers by reducing nuclear factor-kappa B (NF-κB) activation, both in RAW264.7 cells and in an LPS-induced arthritis mouse model. In addition, FP-MD effectively suppressed the expression levels of matrix metalloproteinases (MMPs) at the transcriptional level in inflamed knee joint tissues [27]. However, its effects on the structural joint damage and associated joint pain of OA have not been explored. In this study, we investigated the effects of FP-MD on pain, and the structural changes and inflammatory responses known to cause pain, in a rat model of OA induced by monosodium iodoacetate (MIA) injection.

## 2. Materials and Methods

### 2.1. Preparation of FP-MD

FlexPro MD^®^ (FP-MD) is a commercially available dietary supplement containing a proprietary combination of Superba^®^ Antarctic krill (*Euphausia superba*) oil (Aker BioMarine Antarctic US LLC; Metuchen, NJ, USA), Zanthin^®^ natural astaxanthin derived from *Haematococcus pluvialis*, Flexonic^®^ sodium hyaluronate (the sodium salt of hyaluronic acid) produced from fermentation by *Streptococcus zooepidemicus* (Valensa International, Inc.; Eustis, FL, USA), and food vehicles (Table 1). A sample of FP-MD was produced as reported previously [28,29], then stored at room temperature until use. The FP-MD sample consisted of 70% krill oil, 7% *Haematococcus pluvialis* extract, and 7% sodium hyaluronate, along with 16% various excipients (beeswax, olive oil, etc.), as determined by high-performance liquid chromatography (HPLC) and gas chromatography (GC), using the methods recommended by the United States Pharmacopeia (USP) and Korean Pharmacopoeia (KP), respectively.

### 2.2. Animals and Ethics Statement

Adult male Sprague-Dawley (SD) rats (aged 7 weeks with body weights ranging 200–214.0 g) were obtained from Orientbio Inc. (Sungnam, Gyeonggi, Korea) and housed in the animal facility at Biotoxtech Co. Ltd. (Cheongju, Chungbuk, Korea), a non-clinical Good Laboratory Practice (GLP)-certified Contract Research Organization (CRO). The rats were acclimated for 7 days and maintained under conventional housing conditions at 23 ± 2 °C with a controlled 12 h light/dark cycle, and were provided filtered tap water and a rodent chow diet (Envigo RMS Inc., Indianapolis, IN, USA) *ad libitum* throughout the experiment. All animal experimental procedures complied with the National Institute of Health Guide for the Care and Use of Laboratory Animals and the Korean National Animal Welfare Law. The experimental animal facility and all protocols involving animals in this study were reviewed and approved by the Institutional Animal Care and Use Committee of Biotoxtech (IACUC approval number 180644).

### 2.3. Experimental Design and Administration

The experimental design of the study is shown in Appendix A. Rats were randomly distributed in six groups of eight rats each, consisting of a normal sham control (Sham) group; an MIA-induced OA control (MIA) group; a positive control (PC) group; and three groups treated with increasing doses of FP-MD (25 mg/kg, 50 mg/kg, or 100 mg/kg). All of the groups received treatments via oral gavage once daily for 7 days before OA induction and then for 21 days thereafter until the end of the experiment. The PC group received Celecoxib administered orally at a dose of 3 mg/kg. The Sham and MIA groups received equivalent volumes of vehicle (corn oil). The administration volumes were 5 mL/kg body weight for all experimental groups. The animals were closely monitored for individual clinical signs and other complications by a veterinarian. Body weight was measured weekly, and the weight-bearing levels of the two hind limbs were measured at 0, 3, 7, 14, and 21 days post-OA induction.

### 2.4. MIA-Induced OA Animal Model

Following the 7-day pretreatment period, the animals were anesthetized with 2% isoflurane delivered via a nose cone and then their right hind knees were injected, either with 3 mg MIA (Sigma-Aldrich Inc., St. Louis, MI, USA) in 50 μL saline or an equivalent volume of sterile saline vehicle, through the infrapatellar ligament as previously described [30,31,32] while contralateral knees remained intact. The choice of MIA dose was supported by previous work demonstrating that 3 mg of MIA not only induced joint degeneration but also produced significant axonal injury to dorsal root ganglion cells, reproducing the neuropathic pain component typically observed at the later stages of OA development [33,34]. Following injection, the animals were allowed to fully recover from the anesthesia and were monitored appropriately before returning them to the cages, and then continued to receive the same oral doses of FP-MD, Celecoxib, or vehicle, respectively, for 3 weeks.

### 2.5. Weight-Bearing Test (Pain Assessment)

The weight-bearing distributions between the postoperative and normal hind limbs were determined at 0, 3, 7, 14, and 21 days post-OA induction by using an incapacitance meter (Bioseb Co.; Pinellas Park, FL, USA) as previously described [35,36], which independently measures the weight, in grams, that the animal distributes to each hind paw. Generally, normal animals distribute their weight equally on both hind paws, while animals with MIA-induced joint pain tend to favor the normal limb. The rats were placed in a container with their hind paws comfortably resting on two separate sensor plates. When the rats stand, they make natural adjustments to the weight distribution on both rear paws based on the level of joint pain they are experiencing. The equipment was set to average the weight measured by each sensor over a 5-second period, and a total of three measurements were taken and averaged for each rat. The result was expressed as the mean of weight ratio between injured and non-injured limbs.

### 2.6. Histological Analysis

Histological changes were assessed to determine the effects of FP-MD and Celecoxib on joint cartilage degeneration. After 3 weeks post-OA induction, the rats were euthanized with CO_2_, and the affected knee joints were dissected and fixed in 10% formalin for 24 h at 4 °C, decalcified with 5% formic acid for 2 weeks, and then dehydrated in graded acetone and embedded in paraffin. Sections (thickness, 4–5 μm) were stained with hematoxylin and eosin (H&E) and Safranin-O/Fast green to evaluate structural cartilage damage and the proteoglycan loss, respectively, and then observed under Carl Zeiss Axio Imager A2 microscope (Carl Zeiss, Deisenhofen, Germany). All stained slides were histologically evaluated and statistically graded on a scale of 0–13 by double-blind observation, according to the modified Mankin scoring system [37].

### 2.7. Serum Analysis

Blood samples were collected serially, either from the jugular vein or abdominal vena cava, at 22 days post-OA induction. Blood samples were centrifuged at 3000 rpm for 12 min to separate the serum. The sera were then stored at −70 °C until used for enzyme-linked immunosorbent assay (ELISA) experiments. The serum levels of the pro-inflammatory cytokines tumor necrosis factor alpha (TNF-α), interleukin-1 beta (IL-1β), and interleukin-6 (IL-6), and the cartilage degeneration mediators cartilage oligomeric matrix protein (COMP) and C-telopeptide of type II collagen (CTX-II), were determined using commercial ELISA kits (TNF-α: cat no. BMS622TEM, Invitrogen, MN, USA; IL-1β: cat no. BMS630; IL-6: cat no. BMS625; COMP: cat no. abx256440, Abbexa Ltd., Cambridge, UK; CTX-II: cat no. E-EL-R2554, Elabscience Biotechnology Inc., Houston, TX, USA) according to manufacturer’s protocol, respectively.

### 2.8. Real-Time PCR Analysis

Total RNA was extracted from articular cartilage tissues by using QIAzol^®^ and purified using the miRNeasy Mini Kit (Qiagen GmbH, Hilden, Germany) according to the manufacturer’s protocol. Total RNA was reverse transcripted into complementary DNA (cDNA) using a High Capacity RNA-to-cDNA kit (Applied Biosystems, Foster City, CA, USA), and then subjected to quantitative real-time PCR (qPCR) using QuantiFast^®^ SYBR Green PCR master mix (Qiagen) with custom-designed specific primers using 18S as house-keeping control on a StepOnePlus^TM^ Real-Time PCR System (Applied Biosystems). The primer sequences are listed in Table 2. Relative fold-changes in target gene expression between groups were determined for all targets using the 2^ΔΔCt^ method.

### 2.9. Statistical Analysis

Data were expressed as mean ± standard error of measurement (SEM) for the indicated number of experiments. The statistical significance of the difference between groups was analyzed using independent sample *t*-tests or one-way analysis of variance (ANOVA) followed by Dunnett’s post-hoc test. Statistical analysis was performed using SPSS software version 20.0 (IBM Co.; Armonk, NY, USA). For all tests, *p*-values below 0.05 were considered statistically significant.

## 3. Results

### 3.1. FP-MD Reduces the OA-Induced Joint Pain in Rats

We initially tested the effect of oral administration of FP-MD on joint pain in an MIA-induced rat model of OA. Since OA is accompanied by pain, the severity of pain is mainly determined by the asymmetric weight-bearing distribution in the hind limbs. Figure 1 demonstrates the latency differences between right (ipsilateral) and left (contralateral) hind-paws in the sham control (Sham) and experimental groups, which were analyzed to evaluate OA-induced pain using a capacitance tester for 3 weeks. The MIA-induced OA control (MIA) group animals showed a quick reduction of weight-bearing distribution in the OA-induced limbs compared to non-OA-induced knees due to pain induced by MIA injection, while the weight distribution did not change in the sham control group. In contrast, FP-MD- and Celecoxib-treated group animals showed decreased ipsilateral latency throughout the experimental period as compared with MIA group. Moreover, rats treated with doses of 50 and 100 mg/kg of FP-MD and the Celecoxib-treated rats were able to balance the right and left hind-paws and practically returned to the normal control condition. These data indicate the significant restoration of hind-limb weight bearing in the FP-MD-treated rats. In addition, we found that FP-MD treatment does not affect regular body weight gain in rats (Appendix A).

### 3.2. FP-MD Suppressed Articular Cartilage Damage in MIA-Induced OA Rats

As articular cartilage degeneration is the major histopathological feature of OA joints, the effect of FP-MD administration on the histopathological changes and severity of damage in the articular cartilage were evaluated using H&E and Safranin O-fast green staining in the MIA-induced OA rats. As shown in Figure 2A, the sham control group animals exhibited normal articular cartilage structures with smooth articular surfaces, normal chondrocytes with columnar orientation, and intact tide marks and subchondral bone. In contrast, the MIA group showed the severity of surface irregularity and cleft, matrix loss of articular cartilage, degeneration of columnar orientation, degeneration of the tide mark, and the penetration of subchondral bones. We found that oral administration of FP-MD attenuated the structural morphological changes in the articular cartilage, reduced the penetration of the subchondral bones, and reduced the degeneration of the tide marks in comparison to the MIA group. Proteoglycans as one of the major components of the extra cellular matrix (ECM) have various functions in the cartilage. Safranin O-fast green staining revealed that the joints of the MIA-induced OA rats showed joint space narrowing with marked proteoglycan depletion, whereas sham control rats showed the presence of intense proteoglycan in the ECM. Administration with FP-MD and Celecoxib effectively decreased the loss of proteoglycan in the knee joints compared to MIA group. Furthermore, the severity of OA lesions, graded using the modified Mankin’s scoring system, and the overall modified Mankin’s scores were significantly lower in FP-MD and Celecoxib-treated groups compared with MIA group (Figure 2B).

### 3.3. FP-MD Suppressed the Pro-Inflammatory Cytokine Levels in MIA-Induced OA Rats

The inflammatory response is an important factor associated with OA pathogenesis, and pro-inflammatory cytokines play a prominent role in the maintenance of tissue injury and chronic inflammation during the progression of OA [38]. We, therefore, examined the effect of FP-MD on the production of inflammatory cytokines associated with OA, such as TNF-α, IL-1β, and IL-6, in MIA-induced OA rats. As shown in Figure 3, the MIA group showed increased serum levels of TNF-α, IL-1β, IL-6, and IFN-γ compared with those in the sham control group. In the FP-MD- and Celecoxib-treated groups, the serum levels of pro-inflammatory cytokine were significantly decreased in a dose-dependent manner when compared to the MIA group. These results suggest that FP-MD protects cartilage in the MIA-induced OA model by modifying these pro-inflammatory cytokines.

### 3.4. FP-MD Reduced Biomarkers of Chondrocyte Death in MIA-Induced OA Rats

The effect of FP-MD on the levels of COMP and CTX-II, which are well-established biomarkers for OA diagnosis and progression, which are degradation products of joint tissues, especially the cartilage ECM, during progressive destruction of articular cartilage in OA [39,40], was also investigated. The serum levels of COMP and CTX-II were increased in OAC group by MIA-injection compared with those in the sham control group. In contrast, the serum levels of COMP (Figure 4A) and CTX-II (Figure 4B) were significantly lower in the OA rats treated with FP-MD at doses of 50 and 100 mg/kg, but not 25 mg/kg, than in the OAC group. The MIA-induced overproduction of COMP and CTX-II was also decreased by Celecoxib treatment.

### 3.5. FP-MD Suppressed mRNA Expression of Inflammatory Mediators and Matrix-Degrading Ezzymes in MIA-Induced OA Rats

We next investigated the effect of FP-MD on the mRNA expression levels of inflammatory mediators, inducible nitric oxide synthase (iNOS) and cyclooxygenase-2 (COX-2), and matrix-degrading enzymes, matrix metalloproteinase (MMP)-2 and MMP-9, in the knee joint tissues of MIA-induced OA rats. The mRNA expression of iNOS, COX-2, MMP-2, and MMP-9 increased in OAC group by MIA-injection compared with those in the sham control group (Figure 5). In the FP-MD- and Celecoxib-treated groups, the expression levels of those genes were significantly suppressed. These results suggested that FP-MD inhibited inflammatory responses and articular cartilage damage by inhibiting inflammatory cytokines and MMPs in MIA-induced OA rats.

## 4. Discussion

In the present study, we evaluated whether or not FP-MD exerts an anti-osteoarthritic effect in the MIA-induced OA rat model, which is one of the well-established animal models for human OA and joint pain. In rats, injection of MIA into joints successfully induces acute inflammation, articular cartilage degradation, and joint pain by the direct interruption of chondrocyte metabolism and the subsequent induction of chondrocyte death, representative of the changes observed in patients with OA [41]. Our data show that oral administration of FP-MD led to a significant reduction in MIA-induced joint pain and a decrease in structural changes, including joint space narrowing and cartilage destruction, which were associated with the reduction of pro-inflammatory cytokines and articular cartilage degeneration biomarkers.

Several clinical studies have demonstrated that daily consumption of krill oil (2 g/day for 30 days) improves the subjective symptoms of knee pain in adults with mild knee pain [42], mitigates the subjective symptoms of OA as assessed by Western Ontario and McMaster Universities Osteoarthritis Index (WOMAC), and reduces C-reactive protein (CRP) levels in patients with rheumatoid arthritis or OA with CRP levels greater than 1.0 mg/dL [43]. In addition, the anti-arthritis, anti-pain, and anti-inflammatory effects of krill oil were also observed in a carrageenan-induced mouse model of inflammatory pain [44,45]. Furthermore, current studies have demonstrated that astaxanthin is a promising anti-inflammatory and anti-pain agent against carrageenan-induced paw edema and pain behavior, as well as neuropathic pain [46,47,48]. Hyaluronic acid is also known as a useful treatment option for OA that can modify symptoms and relieve joint pain in OA [49,50,51]. Interestingly, krill oil, astaxanthin, and hyaluronic acid have also been shown to play a role in gut flora modulation, an emerging area of investigation that may also influence the development and progression of osteoarthritis, and symptom management [52,53,54]. Considering the beneficial therapeutic properties of krill oil, astaxanthin, and hyaluronic acid on arthritis, we have developed a novel multi-ingredient dietary supplement formulation, FP-MD, consisting of krill oil, natural astaxanthin, and proprietary lower molecular weight hyaluronic acid, to address the key factors involved in maintaining joint health. In the present study, we performed in vivo animal studies to further investigate the therapeutic potential of FP-MD for pain relief and chondroprotection in an MIA-induced OA rat model by measuring weight-bearing distribution, histopathological changes, serum levels of pro-inflammatory cytokines, and biomarkers for degradation products of articular cartilage.

Here, we demonstrated that oral administration of FP-MD has pain-relieving effects in MIA-induced OA in rats, consistent with our previous findings in a clinical trial that showed remarkable pain relief in subjects suffering from chronic mild-to-moderate knee joint pain [26]. Moreover, histopathological analysis revealed that the FP-MD-treated group exhibited marked suppression of structural changes, bone resorption, and proteoglycan degradation in the MIA-treated rat knee joint. Taken together, these observations suggest that FP-MD possesses potential pain-relieving activity in OA by reducing articular cartilage damage.

The degradation of articular cartilage, including the degradation of cartilage cells and matrix, is the main pathological characteristic of OA, and matrix degeneration mainly results in the losses of proteoglycans and type II collagen [55]. Several previous studies have demonstrated that the serum levels of some structural molecules and fragments derived from articular cartilage, bone, and the synovium, all of which are affected by OA, are elevated in OA patients and OA animal models, thereby they can be used as diagnostic biomarkers which can potentially predict the increased risk and progression of OA [56,57]. COMP and CTX-II are two biochemical markers that are degradation products of joint tissues, especially the cartilage extracellular matrix, and can potentially predict the destruction of articular cartilage in OA. COMP, a tissue-specific pentametric glycoprotein, is one of the essential components of the extracellular matrix of the cartilage which binds to type II collagen fibers and stabilizes the collagen fiber network of articular cartilage in cooperation with other matrix proteins. However, when the articular cartilage is destroyed during the development of OA and under inflammatory conditions, the levels of COMP is noticeably increased in synovial fluid and serum and is positively correlated with joint damage in knee OA [58]. In addition, levels of CTX-II, a degradation product of type II collagen produced by proteases activated by cartilage injury or degeneration, are also elevated in OA patients as compared with normal individuals and are associated with both the prevalence and progression of OA [59].

Accordingly, the serum levels of these two factors, COMP and CTX-II, have the potential to be prognostic biomarkers for monitoring cartilage degradation in patients with OA. Thus, the serum levels of COMP and CTX-II were analyzed to assess the effects of FP-MD on the degradation of articular cartilage. Consistent with previous studies [60,61], serum levels of both COMP and CTX-II were significantly higher in the OAC group than in the sham control group; however, treatment with FP-MD significantly reduced this increase. These findings suggest that the reduction in COMP and CTX-II serum levels by FP-MD most likely represents suppressed MIA-induced degradation of cartilage, as damaged cartilage is a major contributor to circulating COMP and CTX-II levels.

Increasing evidence has demonstrated that the significantly elevated levels of pro-inflammatory cytokines observed in OA patients play a critical role in the promotion of the catabolic processes in OA, causing cartilage degradation [62,63]. High levels of pro-inflammatory cytokines, such as TNF-α, IL-1β, IL-6, and IL-10, have been found in synovial fluid from OA patients and several experimental animal models of cartilage degradation [64]. Among these cytokines, TNF-α and IL-1β are highly overexpressed in the cartilage as well as in the synovial tissue and are considered the major mediators in the OA pathogenesis [65,66]. These cytokines are known to drive the inflammatory cascade, and their increased production induces catabolic events as they downregulate the synthesis of ECM structural components, including proteoglycan, by inhibiting the anabolic activities of chondrocytes and enhancing MMPs, resulting in the loss of cartilage and increased bone resorption during the development and progression of OA [65,67]. IL-6 has also been reported to play a major role in OA. Although the production of IL-6 by chondrocytes is considerably low in physiological conditions, its production can be stimulated by the number of other cytokines and inflammatory mediators, including TNF-α, IL-1β, IFN-γ, and prostaglandin E_2_, resulting in decreased production of type II collagen [68].

Many previous studies have demonstrated that anti-inflammatory agents capable of inhibiting the production of these cytokines might have the potential to control or treat OA [65,66,67]. Hence, we investigated the anti-inflammatory effects of FP-MD by measuring the serum levels of the pro-inflammatory cytokines TNF-α, IL-1β, and IL-6 in MIA-induced OA rats, and found that oral administration of FP-MD effectively decreased the serum levels of these cytokines. In addition, FP-MD significantly reduced the mRNA expression levels of inflammatory mediators, iNOS and COX-2, and matrix-degrading enzymes, MMP-2 and MMP-9, in the knee joint tissue. Taken together, the present results indicate that FP-MD has the potential to blunt inflammatory responses, and which may subsequently reduce articular cartilage damage.

## 5. Conclusions

In conclusion, the present study has demonstrated that oral administration of FP-MD effectively attenuates joint pain and the severity of articular cartilage destruction in an MIA-induced OA rat model and that the anti-osteoarthritic effects of FP-MD were associated with the protection of articular cartilage against inflammation-induced degradation thorough the suppression of pro-inflammatory cytokines. Our findings suggest that FP-MD is a promising dietary supplement for reducing pain, minimizing cartilage damage, and improving functional status in OA patients, and could overcome the disadvantages of previous dietary supplements, including glucosamine and chondroitin sulfate, as well as medicinal agents, such as corticosteroids and NSAIDs.

## Figures and Tables

**Figure 1 nutrients-12-00956-f001:**
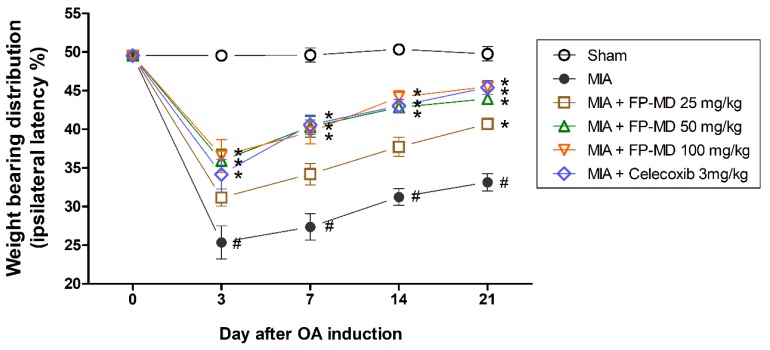
Effects of oral administration of FlexPro MD^®^ (FP-MD) on changes in hind-paws weight-bearing distribution in monosodium iodoacetate (MIA)-induced osteoarthritis (OA) rats. The weight-bearing distribution ratio was measured for 21 days after injection of MIA using an incapacitance tester, compared to that of the vehicle-treated MIA-induced group. Data are expressed as the mean ± S.E.M (*n* = 8). # *p* < 0.05 versus vehicle-treated Sham group and * *p* < 0.05 versus MIA.

**Figure 2 nutrients-12-00956-f002:**
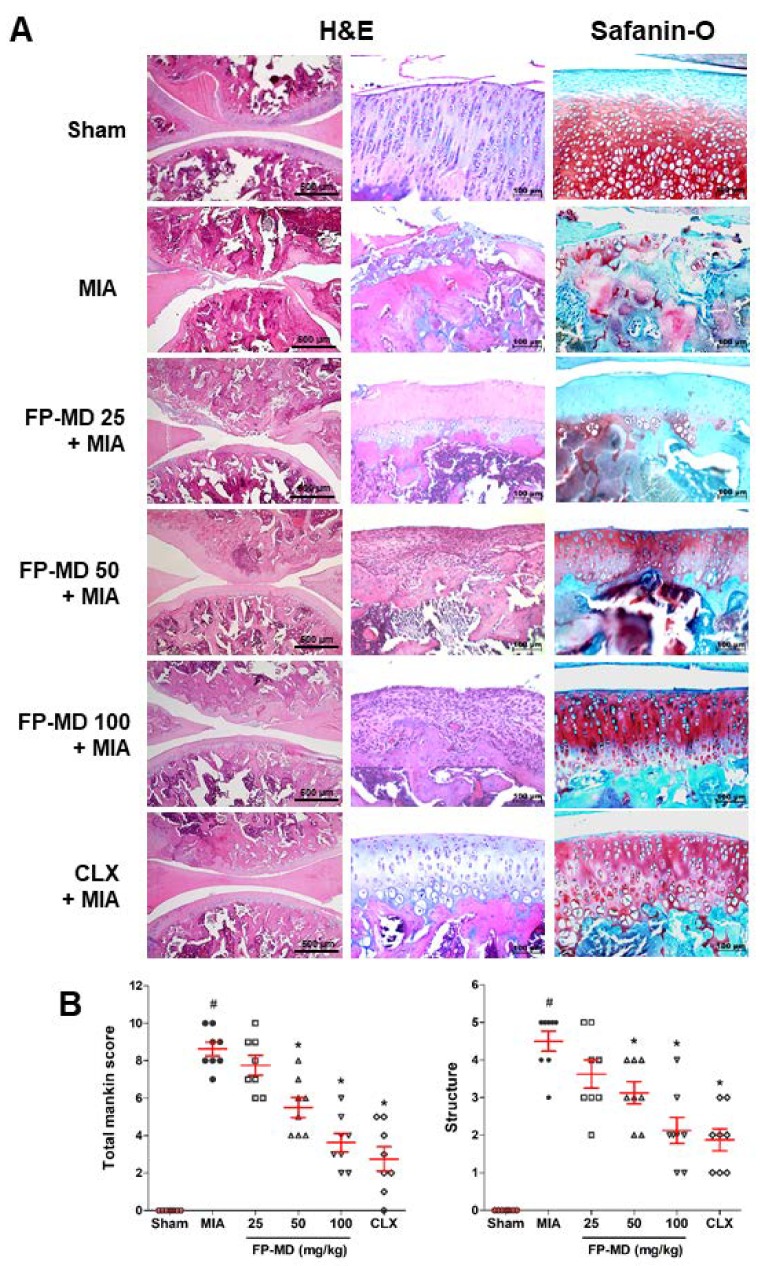
Histological evaluation of joints activity after administration with FlexPro MD^®^ (FP-MD) in monosodium iodoacetate (MIA)-induced osteoarthritis (OA) rats. (**A**) Knee joints of the OA rats were stained with hematoxylin and eosin (H&E) and Safranin O-Fast green. (**B**) The joint lesions were graded on a scale of 0–13 using the modified Mankin’s scoring system, giving a combined score for cartilage structure. Data are expressed as the mean ± S.E.M (*n* = 8). # *p* < 0.05 versus vehicle-treated Sham group and * *p* < 0.05 versus MIA. CLX; celecoxib 3 mg/kg-treated.

**Figure 3 nutrients-12-00956-f003:**
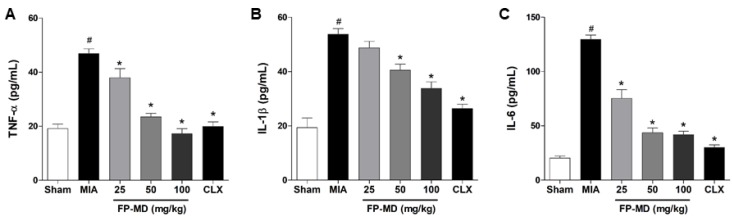
Effects of oral administration of FlexPro MD^®^ (FP-MD) on the level of tumor necrosis factor alpha (TNF-α), interleukin (IL)-1β, and IL-6 in monosodium iodoacetate (MIA)-induced osteoarthritis (OA) rats. The serum concentrations of (**A**) TNF-α, (**B**) IL-1β, and (**C**) IL-6 in FP-MD (25–100 mg/kg) + MIA- or Celecoxib (CLX) (3 mg/kg) + MIA-induced OA rats were compared to those of the vehicle-treated MIA group. Data are expressed as the mean ± S.E.M (*n* = 8). # *p* < 0.05 versus vehicle-treated Sham group and * *p* < 0.05 versus MIA.

**Figure 4 nutrients-12-00956-f004:**
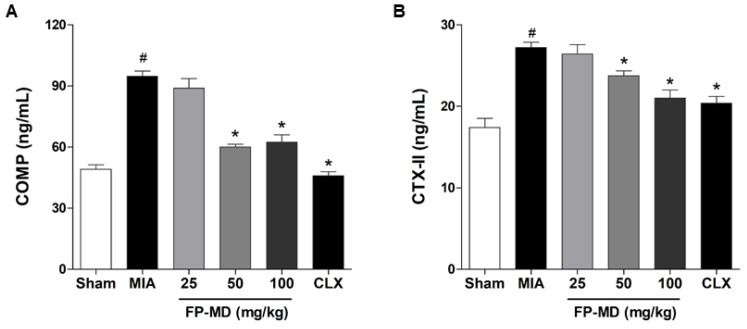
Effects of oral administration of FlexPro MD^®^ (FP-MD) on the production of cartilage oligomeric matrix protein (COMP) and C-telopeptide of type II collagen (CTX-II) in monosodium iodoacetate (MIA)-induced osteoarthritis (OA) rats. The serum concentrations of (**A**) COMP and (**B**) CTX-II in FP-MD (25–100 mg/kg) + MIA- or Celecoxib (CLX) (3 mg/kg) + MIA-induced OA rats were compared to those of the vehicle-treated MIA group. Data are expressed as the mean ± S.E.M (*n* = 8). # *p* < 0.05 versus vehicle-treated Sham group and * *p* < 0.05 versus MIA.

**Figure 5 nutrients-12-00956-f005:**
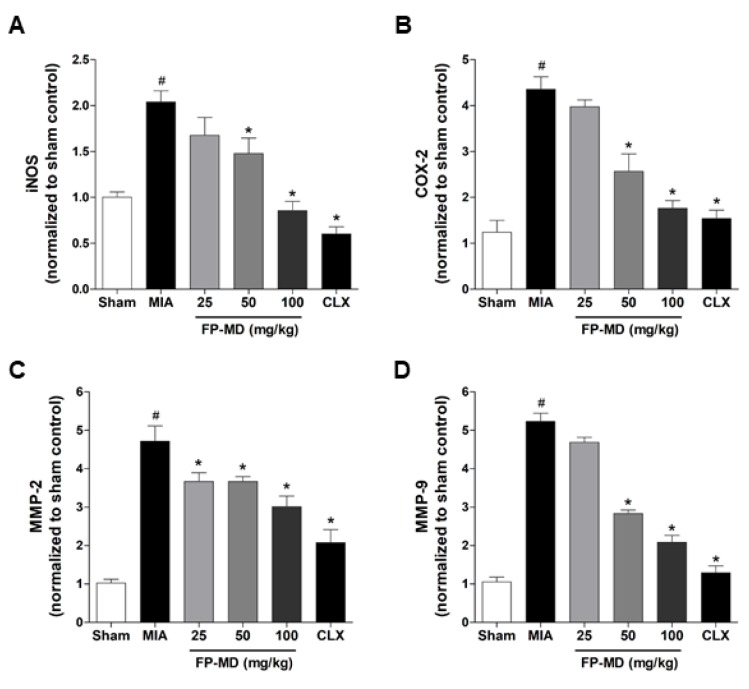
Effects of oral administration of FlexPro MD^®^ (FP-MD) on the expression of inflammatory mediators and metalloproteinases (MMPs) in the knee joint in monosodium iodoacetate (MIA)-induced osteoarthritis (OA) rats. The mRNA expression of (**A**) inducible nitric oxide synthase (iNOS), (**B**) cyclooxygenase-2 (COX-2), (**C**) MMP-2, and (**D**) MMP-9 in FP-MD (25–100 mg/kg) + MIA- or Celecoxib (CLX) (3 mg/kg) + MIA-induced OA rats were compared to those of the vehicle-treated MIA group. Data are expressed as the mean ± S.E.M (*n* = 8). # *p* < 0.05 versus vehicle-treated Sham group and * *p* < 0.05 vs. MIA.

**Table 1 nutrients-12-00956-t001:** Composition of FlexPro MD^®.^

Ingredient	Amount (mg)	Ratio (%)
Antarctic krill oil	321	70
*Haematococcus pluvialis* extract (to deliver 2 mg astaxanthin)	25–35	5.5–7.5
Sodium hyaluronate	33	7.1
Excipients	73–83 ^1^	15.4–17.4
Total	462	100

^1^ adjust the amount depending on the amount of *Haematococcus pluvialis* extract.

**Table 2 nutrients-12-00956-t002:** List and sequences of real-time PCR primers for mRNA expression.

Gene	Primer Sequence	
iNOS	Forward	5′-CTTTACGCCACTAACAGTGGCA-3′
	Reverse	5′-AGTCATGCTTCCCATCGCTC-3′
COX-2	Forward	5′-CCTCGTCCAGATGCTATCTTTG-3′
	Reverse	5′-GAAGGTCGTAGGTTTCCAGTATT-3′
MMP-2	Forward	5′-CACCAAGAACTTCCGACTATCC-3′
	Reverse	5′-TCCAGTACCAGTGTCAGTATCA-3′
MMP-9	Forward	5′-CCCAACCTTTACCAGCTACTC-3′
	Reverse	5′-GTCAGAACCGACCCTACAAAG-3′

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
