# Peer review of "FlexPro MD®, a Combination of Krill Oil, Astaxanthin and Hyaluronic Acid, Reduces Pain Behavior and Inhibits Inflammatory Response in Monosodium Iodoacetate-Induced Osteoarthritis in Rats"

_nutrients, 2020, doi:10.3390/nu12040956_

Round 1

Reviewer 1 Report

Dear Authors,

in this review, you aim to present the use of dietary supplements and nutraceuticals in an effort to improve symptoms and manage progression of osteoarthritis.

The topic is really interesting and the article is well written.

On the other hand, I suggest some minor revisions to improve this work.

Minor revisions

Introduction and Discussion

I suggest to describe the emerging concept of “gut-joint axis” and explain the role of diet on the gut microbiota modulation in osteoarthritis. Additional references recommended:

  • Gut-joint axis: the role of physical exercise on gut microbiota modulation in older people with osteoarthritis (DOI:10.3390/nu12020574)
  • The gut microbiome-joint connection implications in osteoarthritis (DOI:10.1097/BOR.0000000000000681)
  • Cartilage-gut-microbiome axis: a new paradigm for novel therapeutic opportunities in osteoarthritis (DOI:10.1136/rmdopen-2019-001037)

Author Response

Comment: In this review, you aim to present the use of dietary supplements and nutraceuticals in an effort to improve symptoms and manage progression of osteoarthritis. The topic is really interesting and the article is well written. On the other hand, I suggest some minor revisions to improve this work.

I suggest to describe the emerging concept of “gut-joint axis” and explain the role of diet on the gut microbiota modulation in osteoarthritis. Additional references recommended:

Gut-joint axis: the role of physical exercise on gut microbiota modulation in older people with osteoarthritis (DOI:10.3390/nu12020574)

The gut microbiome-joint connection implications in osteoarthritis (DOI:10.1097/BOR.0000000000000681)

Cartilage-gut-microbiome axis: a new paradigm for novel therapeutic opportunities in osteoarthritis (DOI:10.1136/rmdopen-2019-001037)

Author’s Response:
We thank the reviewer for this comment. To address your comment, we included some background information about the emerging concept of the “gut-joint axis” to the Introduction section in the revised version of the manuscript. And, we have further added references to the Discussion that show the relevance of the “gut-joint axis” as an emerging field relevant to the ingredients in FP-MD.

Reviewer 2 Report

The paper analyzes the effects of FP-MD, a multi-ingredient dietary supplement, on pain and histological damage in a rat model of osteoarthrithis induced by MIA. The aim of the study is concise and clear, the methods are appropriate to achieve the proposed objectives and results obtained are interesting and conclusive.  

Major points:

Considering that the protective effects of this compound had already been reported in an LPS-induced artrithis mouse model, it would of great interest to delve deeper into the mechanism of action of this compound in the MIA rat model.

To analyze the effects of the compound on the phosphorilation of NFkB and the mRNA expression of pro-inflammatory enzymes such as iNOS or COX-2 and the protein expression of MMP13, would significantly shed a light on the mechanism of action of this compound.

It is necessary to significantly reduce the discussion of the manuscript, trying to focus on the results obtained in the study and how these results fit with the literature reported (From line 250 to 296, the discussion makes no reference to results of the present manuscript).  

Author Response

Comment 1: Considering that the protective effects of this compound had already been reported in an LPS-induced arthritis mouse model, it would of great interest to delve deeper into the mechanism of action of this compound in the MIA rat model. To analyze the effects of the compound on the phosphorylation of NF-kB and the mRNA expression of pro-inflammatory enzymes such as iNOS or COX-2 and the protein expression of MMP13, would significantly shed a light on the mechanism of action of this compound.

Author’s Response:
We thank the reviewer for this suggestion. To address your comment, we analyzed the effect of FP-MD on the expression of pro-inflammatory genes (iNOS and COX-2) and matrix metalloproteinases (MMP-2, MMP-9, and MMP-13) by performing qRT-PCR analysis. We observed increased expression levels of both iNOS and COX-2 in the cartilage of the MIA-induced OA rats, whereas FP-MD-treated rats showed reduced expression. In addition, FP-MD treatment effectively reduced MMP-2 and MMP-9 mRNA expression. However, somehow we did not observe increased MMP-13 expression in our experimental conditions. Thus, we now have mRNA expression data in the revised manuscript (see Fig. 5). These results suggested that FP-MD inhibits the inflammatory response and cartilage ECM degradation by suppressing the expression of pro-inflammatory genes and matrix metalloproteases. Unfortunately, we do not have protein samples isolated from joint cartilages. Therefore, we could not include phosphorylation of NF-kB and MMP-13 protein expression data that you suggested in the revised manuscript. Nevertheless, we thank you for the reviewer’s insightful suggestion, and we believe that our experimental evidence from multiple in vivo functional tests, pro-inflammatory cytokine levels, and chondrocyte death biomarkers would be strong enough to present our research findings to readers even though we could not demonstrate protein expression test results. We thus hope the reviewer would agree on the relevance of our study and the novel of our findings merit publication.

Comment 2: It is necessary to significantly reduce the discussion of the manuscript, trying to focus on the results obtained in the study and how these results fit with the literature reported (From line 250 to 296, the discussion makes no reference to results of the present manuscript).

Author’s Response: The manuscript was modified accordingly.

Reviewer 3 Report

This study used a commercial dietary supplement FlexProMD® composed of krill oil, astaxanthin, and hyaluronic acid to evaluate its effects on osteoarthritis (OA) progress in a rat model with monosodium iodoacetate-induced OA. The results showed that FlexProMD treatment might significantly reduce serum cartilage degeneration biomarkers, such as cartilage oligomeric matrix protein (COMP), crosslinked C-telopeptide of type II collagen (CTX-II), and proinflammatory cytokines of tumor necrosis factor α (TNF -α), interleukin 1β (IL-1β) and interleukin 6 (IL-6). Accordingly, authors concluded that “FP-MD is a promising dietary supplement for reducing pain, minimizing cartilage damage,…..”.

This study aimed to reduce OA progress by dietary supplementation, however, there were not enough scientific evidences to support their conclusion. In addition, there are too many inappropriate and non-scientific words used, such as how to evaluate pain degree in a rat model; how to evaluate minimizing cartilage damage. There are still more points should be explained and discussed.. Example:  

  1. In the experimental animal design, rats were randomly divided into six groups, with eight rats in each group. Did the sex be concerned?.
  2. Functions of pro-inflammatory cytokines of TNF -α, IL-1β and IL-6 have well known and published in many papers. What are signals in this study?.
  3. What is the effect of anti-OA progress in each FP-MD ingredient?

The results do not strongly support authors’ conclusion.

Author Response

Comment 1: In the experimental animal design, rats were randomly divided into six groups, with eight rats in each group. Did the sex be concerned?

Author’s Response:
In the current study, we employed the randomization method for grouping that is well known as a gold-standard methodology, and fundamental to designing good experiments as it minimizes the chance of a biased result (Hirst et al., The need for randomization in animal trials: an overview of systematic reviews., PLoS One 2014: 6(6):e98856). In addition, male SD rats (aged seven weeks with 200-214.0 g body weight range) were studied in our current manuscript, as we described in Method section 2.2. Animals and ethics statement. The primary reason for choosing male animals was that recent literature has noted the effect of sex (hormonal status) as well as age, on the results as possible limitations (Kuyinu et al., Animal models of osteoarthritis: classification, update, and measurement of outcomes. J Orthop Surg Res 2016; 11:19). Moreover, it is well known that the sex of the animal (sex hormones) plays a critical role in the progression of OA in the murine model, with males having more severe OA than females. Intact females had more OA than OVX females, indicating that ovarian hormones decrease the severity of OA in the female animal (Ma et al., Osteoarthritis severity is sex-dependent in a surgical mouse model., Osteoarthritis Cartilage 2007; 15(6):695-700). With this background, we used male animals only in the current study, and thus our findings do not present any evidence for sex differences.

Comment 2: Functions of pro-inflammatory cytokines of TNF -α, IL-1β and IL-6 have well known and published in many papers. What are the signals in this study?

Author’s Response:
We appreciate your comment. As we described in the Discussion section, many studies reported that significantly elevated levels of pro-inflammatory cytokines, including TNF-α, IL-1β, and IL-6 observed in OA patients, play a critical role in the promotion of the catabolic processes in OA, causing cartilage degradation. Moreover, these pro-inflammatory cytokines are known to drive the inflammatory cascade, and their increased production induces catabolic events as they downregulate the synthesis of ECM structural components, including proteoglycan, by inhibiting the anabolic activities of chondrocytes and enhancing MMPs, resulting in the loss of cartilage and increased bone resorption during the development and progression of OA. Many previous studies have demonstrated that anti-inflammatory agents capable of inhibiting the production of these cytokines might have the potential to control or treat OA. In the current study, our results showed that oral administration of FP-MD effectively decreased the levels of these cytokines. In addition, we observe increased expression levels of pro-inflammatory enzymes (iNOS and COX-2) and matrix metalloproteases in the cartilage of the MIA-induced OA rats, whereas FP-MD-treated rats showed reduced expression (data shown in the revised version of manuscript). Therefore, our findings indicate that FP-MD has the potential to blunt inflammatory responses and ECM degradation, which may subsequently reduce articular cartilage damage and, consequently, pain.

Comment 3: What is the effect of anti-OA progress in each FP-MD ingredient? The results do not strongly support the authors’ conclusion.

Author’s Response:
We are grateful to you for raising this important issue, and appreciate your critical comment. The major ingredients in FP-MD are Krill oil, astaxanthin, and hyaluronic acid. It has been shown to markedly reduce pain in OA patients suffering from chronic mild-to-moderate knee joint pain. As we mentioned in the Discussion section, it has been reported that consumption of Krill oil (2 g/day for 30 days), a major ingredient of FP-MD, mitigates the symptoms of OA, such as knee pain, in OA patients. In addition, the anti-arthritis, anti-pain, and anti-inflammatory effects of Krill oil were also found in a carrageenan-induced mouse model study. Furthermore, Astaxanthin also has been reported as a promising anti-inflammatory and anti-pain agent against carrageenan-induced paw edema and pain behavior, as well as for neuropathic pain. Hyaluronic acid is also known as a useful ingredient for OA pain. Considering the beneficial therapeutic properties of these three ingredients on arthritis, we thus have developed a novel multi-ingredient dietary supplement formulation, FP-MD, to address the key factors involved in maintaining joint health. The purpose of the current study was to investigate the therapeutic potential of FP-MD formulation for pain relief and chondroprotection in an MIA-induced OA rat model rather than identify the effect of each ingredient. Therefore, we did not include experiments to evaluate the anti-OA effect of each individual ingredient. Nevertheless, we thank you for your insightful comment, and we believe that our experimental evidence from multiple in vivo functional tests on OA-mediated pain, inflammatory responses, and cartilage damage would be strong enough to present our research findings to readers even though we could not demonstrate the anti-OA effect of each ingredient. We thus hope the reviewer would agree on the relevance of our study and the novel of our findings merit publication.

Reviewer 4 Report

This an interesting report on a preventive effect of FP-MD in a rat model of OA induced by monosodium iodoacetate (MIA) suggesting reducing inflammation and cartilage destruction.

May I suggest considering the following issues:

  1. FP-MD: The exact composition with a prevalence of krill oil should be given in a table form. The authors might discuss the contribution to the pharmacological effect of the other components as krill oil is known to be active.
  2. Methods: Please give a lit reference for the weight-bearing test, which is used apparently to assess pain/hypernociception.
  3. Graphs: Please present the data throughout as single dot plots to better appreciate single data points.
  4. Microscopic pictures of OA should be of better quality and devoid of artifact as the chondrocyte damage is not well appreciated. Further, the MIA control the inflammation is not visible.
  5. In the discussion, avoid statements that FP-MD is better tolerated, as you did not test that.
  6. It would be of interest to discuss the potential therapeutic effect FP-MD

Author Response

Comment 1: FP-MD: The exact composition with a prevalence of krill oil should be given in a table form. The authors might discuss the contribution to the pharmacological effect of the other components as krill oil is known to be active.

Author’s Response: The manuscript was modified accordingly with a table (Table 1). The major ingredients in FP-MD are Krill oil, astaxanthin, and hyaluronic acid. It has been shown to markedly reduce pain in OA patients suffering from chronic mild-to-moderate knee joint pain. As we mentioned in the Discussion section, it has been reported that consumption of Krill oil (2 g/day for 30 days), a major ingredient of FP-MD, mitigates the symptoms of OA, such as knee pain, in OA patients. In addition, the anti-arthritis, anti-pain, and anti-inflammatory effects of Krill oil were also found in a carrageenan-induced mouse model study. Furthermore, Astaxanthin also has been reported as a promising anti-inflammatory and anti-pain agent against carrageenan-induced paw edema and pain behavior, as well as for neuropathic pain. Hyaluronic acid is also known as a useful ingredient for OA pain. Considering the beneficial therapeutic properties of these three ingredients on arthritis, we thus have developed a novel multi-ingredient dietary supplement formulation, FP-MD, to address the key factors involved in maintaining joint health.

Comment 2: Methods: Please give a lit reference for the weight-bearing test, which is used apparently to assess pain/hypernociception.
Author’s Response: We agree and have now references for the pain asseess method in the revised version of the manuscript.

Comment 3: Graphs: Please present the data throughout as single dot plots to better appreciate single data points. Author’s Response: We appreciate your comment. We modified graphs as single dot plots accordingly in Figure 2B, which presents the individual scores of each animal. However, we would like to keep the graph style of leftovers to provide easy understanding by readers.

Comment 4: Microscopic pictures of OA should be of better quality and devoid of the artifact as the chondrocyte damage is not well appreciated. Further, the MIA control of the inflammation is not visible.

Author’s Response: We have improved our figures' quality in Figure 2A by providing better resolution pictures.

Comment 5: In the discussion, avoid statements that FP-MD is better tolerated, as you did not test that. It would be of interest to discuss the potential therapeutic effect FP-MD.

Author’s Response: We agree and the manuscript was modified accordingly.

Round 2

Reviewer 2 Report

Previous queries have been adressed by authors